# Unveiling hidden rocky reefs of the Mexican Atlantic coast: Topographic characterization and benthic community dynamics along the North Coast of the Yucatan Peninsula, Mexico

Ileana Ortegón-Aznar[1]*, Johnny Omar Valdez-Iuit ᴵᴰ[2]ᵒ*, Armin N. Tuz-Sulub[1]ᵒ, Alain Duran[3]ᵒ

**1** Campus de Ciencias Biológicas y Agropecuarias, Universidad Autónoma de Yucatán (UADY), Mérida, Yucatán, Mexico, **2** Unidad Multidisciplinaria de Docencia e Investigación-Sisal (UMDI-Sisal), Universidad Nacional Autónoma de México (UNAM), Sisal, Yucatán, Mexico, **3** Department of Biological Sciences, Institute of Environment, Florida International University Florida International University, United States of America

ᵒ These authors contributed equally to this work
* jvaldezi@ciencias.unam.mx (VIJ), oaznar@correo.uady.mx (OAI)

## Abstract

Rocky reefs are vital ecosystems hosting diverse benthic communities, including sessile organisms like algae, octocorals, and sponges, alongside associated fish and invertebrates, providing numerous ecosystem services. Despite their ecological significance, rocky reefs along the Mexican Atlantic coast remain understudied. This study presents a comprehensive topographic and ecological analysis of four rocky reef sites along the Yucatan Peninsula: Dzilam (La Poza and Small Mountain Range), Telchac, Progreso, and Chicxulub. Using bathymetric surveys, reef structures were mapped, and underwater surveys analyzed the benthic composition. Bathymetric surveys revealed distinct structural morphologies and rugosity indices, while sediment analysis identified varying grain sizes influencing benthic community composition. Macroalgae dominated the benthic cover (64%), followed by long sediment-laden algal turf-(LSAT, 21%) and sessile invertebrates (e.g., sponges). A strong negative correlation (r = −0.754, P < 0.0001) between macroalgal abundance and LSAT highlights competitive dynamics, modulated by environmental factors such as depth, sediment type, and topographic complexity. This study is the first report of the presence of rocky reefs in Yucatan and Mexican Atlantic coast, these findings underscore the ecological significance of rocky reefs as biodiversity hotspots and their sensitivity to substrate characteristics. This study highlights the need for further spatiotemporal research to understand their ecological dynamics and inform conservation strategies.

**Data availability statement:** All relevant data are within the manuscript and its Supporting Information files.

**Funding:** The author(s) received no specific funding for this work.

**Competing interests:** The authors have declared that no competing interests exist.

## Introduction

Rocky reefs serve as vital marine ecosystems, nurturing a diverse array of benthic life—including sessile forms like macroalgae, octocorals, and sponges—alongside mobile species such as fish and invertebrates [1,2].

These habitats provide essential ecosystem services, such as larval dispersal, breeding grounds, refuge, feeding areas, and support for commercially valuable species [3–5]. In Mexico, reef ecosystems span approximately 1,780 km² and are classified into coral, rocky, mixed, and artificial types [6]. While coral reefs, such as those in the Mesoamerican Reef and Campeche Bank, have garnered substantial research attention [7,8], rocky reefs, despite covering a larger area and exhibiting significant ecological productivity [1], remain underrepresented, particularly along the Yucatan Peninsula.

In the Mexican Atlantic, reef ecosystems are divided into three regions: the Reef Corridor of the Southwest Gulf of Mexico (RCSGM), the Mexican Caribbean segment of the Mesoamerican Reef System, and the Yucatan and Campeche Bank, all of them recognized as coral reefs [5,7]. These studies have largely overlooked rocky formations along the Yucatan coast, despite anecdotal evidence from local fishermen highlighting their presence and importance as fishing grounds. In contrast, rocky reefs along Mexico's Pacific coast, such as in the Gulf of California and Baja California, are well-documented [5,6]. The northern Yucatan's geological context, situated atop a carbonate platform potentially sculpted by the Chicxulub impact crater with a 180 km-wide structure formed 66 million years ago [9–12], may have a unique geological origin tied to this event. Buried under ~1 km of carbonate sediments and marked by a semi-circular ring of cenotes [13], the crater's influence on regional substrate composition suggests these reefs could be remnants or expressions of impact-related processes. However, prior studies in the region have focused predominantly on unconsolidated sandy deposits and submarine dunes [14,15], leaving consolidated rocky formations largely unexamined.

Ecological research on Yucatan's submerged coral reefs, particularly within the Campeche Bank, remains limited. Although marine invertebrates rank among the most biodiverse and abundant groups in global aquatic ecosystems, their study in this region is sparse. Exceptions include localized reports of invertebrate richness [16,17], Gulf of Mexico anemone studies [18], sponge diversity analyses [19,20], and cryptic invertebrate surveys [21]. Macroalgal research includes morphofunctional group analyses by Ortegón-Aznar et al. (2008) [22], coral reef studies by González-Solís et al. (2018) [23], and island benthic algal studies [24,25]. Despite these efforts, comprehensive data on the northern Yucatan's rocky reef communities are lacking.

Despite their ecological importance, rocky reefs along the Yucatan coast face growing threats from coastal development, pollution, overfishing, unregulated anchoring, and erosive degradation [26], while climate change introduces additional stressors such as rising sea temperatures, ocean acidification, and altered current regimes [27–29]. These pressures can disrupt benthic community structure, reduce biodiversity, and compromise ecosystem services [30,31]. Yet, rocky reefs remain largely absent from regional conservation frameworks, which have traditionally focused on

coral reef systems [6]. Documenting and understanding these habitats is therefore essential for inclusive marine spatial planning and long-term ecological resilience

This study aims to characterize the topography and benthic community composition of rocky reef sites along the northern coast of the Yucatan Peninsula using bathymetric surveys and underwater ecological assessments, providing baseline data to explore their ecological role as critical hubs of biodiversity and ecosystem function.

## Materials and methods

### Study area

This study was carried out along the northern coast of Yucatan State, Mexico, a region characterized by its karstic geology, shaped by prolonged submersion during the Cretaceous and Tertiary periods [32]. The Yucatan Peninsula features an expansive continental shelf, stretching 360 km wide and averaging 50 m in depth, situated at the intersection of the Gulf of Mexico and the Caribbean Sea, connected via the Yucatan Channel [33]. The coastal marine environment is defined by mild wave action, predominant east-to-west surface currents, and a climate with three distinct seasons: dry (February–May), rainy (June–October), and "nortes" (November–January), characterized by strong northerly winds [34]. Annual water temperatures range from 24°C to 28°C, with Gulf salinity averaging 36.5 mg/L [35].

Four sampling localities were selected based on traditional ecological knowledge from local fishermen, who identified these locations as productive rocky reef fishing zones: Progreso (21.43000°N, −89.74931°W), Chicxulub (21.48339°N, −89.62892°W), Telchac (21.43486°N, −89.45350°W), and Dzilam (21.57997°N, −88.84883°W) (Fig 1). These localities are situated within the 10–15 m depth contour, approximately 13–20 km offshore.

### Methodology

**Bathymetric surveys.** Rocky reef locations were identified through collaboration with local fishermen, who provided central coordinates for each locality based on their fishing experience. At each location (Progreso, Chicxulub, Telchac, and Dzilam) a polygonal study area was delineated, with the number of transects proportional to the reef's surface area, equidistant at 20 m to capture the greatest possible variability.

Bathymetric data were collected via a Garmin echoMap 52dv multifrequency echosounder (50/200 kHz with ClearVü) featuring built-in GPS. Depth and position data (X, Y, Z coordinates) were logged every second and saved to a high-speed SD card. Raw data underwent error removal in Excel, then were analyzed in Surfer via simple kriging (for analysis and mapping), yielding precise topographic depictions of each reef's structure.

**Environmental and seabed characterization.** Environmental parameters were measured at each locality using a YSI Pro 20–30 multiparameter probe, recording salinity, pH, and temperature were taken on the surface, to establish a physicochemical baseline. Seabed characteristics, including rugosity, depth, and sediment type, were systematically assessed.

Topographic complexity was assessed using the Rugosity Index (R.I.), a widely used field method for estimating structural complexity in benthic habitats. We employed the chain-and-tape technique, in which a 10-meter chain with ~1 cm links was laid along the four benthic transect in each reef surface, following its natural contours,. The linear distance spanned by the chain was measured, and rugosity was calculated as the ratio of chain length to linear distance (R.I. = 1 indicates a flat substrate; lower values reflect greater complexity).

This method has been extensively applied in coral reef and rocky reef studies due to its simplicity, cost-effectiveness, and ability to capture fine-scale surface variation [36,37]. While it does not capture multi-scale topographic variation as precisely as 3D photogrammetry or LiDAR-based approaches, it remains a robust proxy for habitat complexity in field-based ecological assessments.. Sediment depth was measured in millimeters using a vernier caliper inserted vertically until it reached the underlying hard substrate. Sediment samples (four at each site) were collected within a 5 cm² quadrat, and grain size was classified using Folk's (1954) textural categories: very fine sand (0.0625–0.125 mm), fine sand

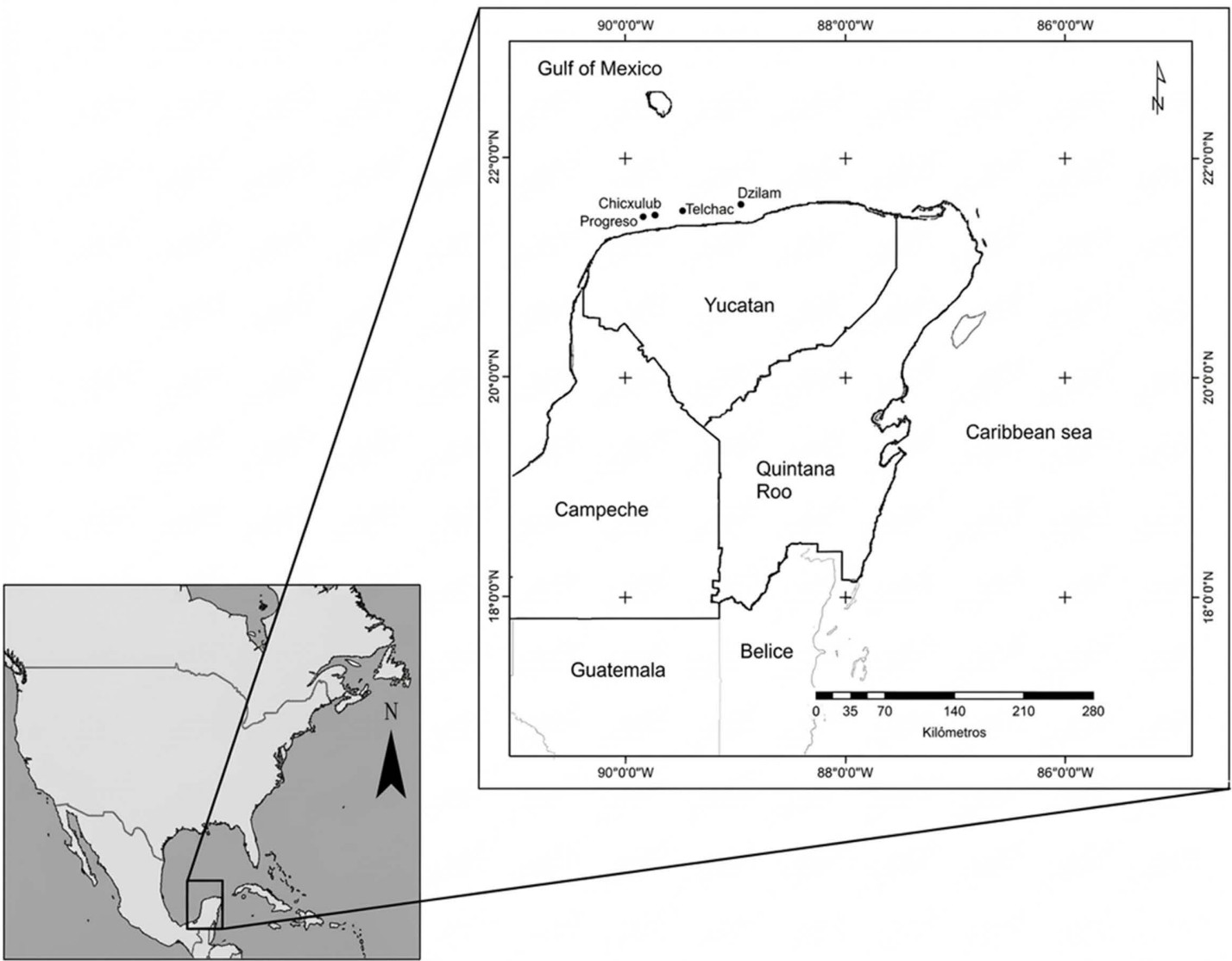

**Fig 1. Map of sampling sites of rocky reefs along the northern Yucatan coast.**

(0.125–0.25 mm), medium sand (0.25–0.5 mm), coarse sand (0.5–1.0 mm), and very coarse sand (1.0–2.0 mm) [38]. This classification elucidated substrate composition and its potential influence on benthic ecology.

**Benthic sampling.** Benthic surveys were conducted seasonally to capture temporal variability, with sampling during the rainy season (July–September 2023 and July 2024) and dry season (May–June 2024). Benthic sampling was conducted via SCUBA diving, employing four transects (20 m) at each locality using 2 sites per reef. Surveys of reefs involved examining 25 plots measuring 25 x 25 cm per site, with plot selection randomized within our designated study areas. Each plot was photographed using an underwater camera (Olympus Tough TG-7 OM SYSTEM) for subsequent analysis. Benthic composition was quantified for relative abundance using Coral Point Count with Excel extensions (CPCe) software (version 4.1) [39], with minor code adjustments. Organisms were categorized into three main groups: (1) Macroalgae, encompassing all algae exceeding 2.5 cm in height [40]; (2) Long Sediment-Laden Algal Turfs (LSAT),

turfs that are laden with sediments forming a layer ~>5mm [41]; and (3) Sessile Invertebrates, including hard corals, fleshy corals, and sponges grouped together due to their ecological similarity as substrate competitors. Relative abundances were calculated for each category, providing insights into community structure and seasonal dynamics.

Ethics statement and permits: This study did not involve the collection of organisms, or any procedures requiring direct interaction with individuals. All benthic surveys were conducted through non-invasive photographic documentation of reef plots. Regarding fieldwork permissions, no collection permits were required, as the research involved only in situ photography of benthic communities without any physical sampling or disturbance. Field site access was conducted in accordance with local regulations, and the study was carried out in publicly accessible marine areas.

The map and all images and photos were made and taken by the authors, and we give consent for their use in the manuscript.

**Statistical analysis.** Statistical analyses were performed using STATGRAPHICS Centurion 19 to evaluate ecological patterns across reef sites and seasonal conditions. A combination of descriptive, inferential, and multivariate approaches was applied. Summary statistics (mean and standard deviation) were calculated for key environmental and biological variables, including rugosity index (R.I.), sediment grain size, depth, LSAT sediment depth, and benthic cover categories such as macroalgae, LSAT, and sessile invertebrates.

To evaluate spatial and temporal variation in benthic composition, two-way ANOVAs were conducted with reef site and season as fixed factors. Separate models were developed for macroalgal cover, LSAT, and sessile invertebrates. When significant effects were observed, Tukey HSD post-hoc tests were used to identify specific differences between reef-season combinations. Interaction plots were generated to visualize these trends.

Pearson correlation analysis was employed to examine relationships among benthic cover, rugosity, sediment depth, and substrate type. A correlation matrix was constructed to summarize statistically significant associations, with thresholds set at $p < 0.05$ and $p < 0.01$.

To determine the environmental drivers of macroalgal abundance, a multiple linear regression model was fitted using macroalgal cover as the dependent variable. Predictor variables included rugosity index, LSAT cover, LSAT sediment depth, reef depth, season, and reef site. Model performance was assessed using $R^2$ and adjusted $R^2$ values, and partial effects were evaluated through standardized regression coefficients and associated p-values.

Finally, bar plots with error bars (± standard error) were created to depict benthic composition across reef sites and seasons, enhancing the clarity of visual comparisons.

## Results

### Topographic characterization of Yucatan rocky reefs

**Dzilam Reef: La Poza.** The La Poza reef (Fig 2a in S1 Fig), located 20.3 km offshore from Dzilam de Bravo, spans a 600 m (E-W) x 200 m (N-S) polygon with 30 N-S transects. Bathymetric data revealed a single rocky structure rising 2 m above the seabed, with depths ranging from −11.6 m (reef surface) to −13.9 m (base) (Fig 2b). The structure features deep caves, enhancing habitat complexity (Fig 2c). The R.I. averaged 0.91 ± 0.35, and sediment was classified as coarse sand (0.725 ± 0.22 mm) (Table 1).

**Dzilam Reef: Small Mountain Range (SMR).** The SMR (Fig 3a in S2 Fig.) spans a 200 m (E-W) x 400 m (N-S) polygon with 22 E-W transects. It features a slab-like formation extending 350 m N-S, with depths decreasing from −6.31 m (north) to −5.06 m (south) (Fig 3b). The maximum elevation is 1.2 m, with potential northward extension (Fig 3c). The R.I. averaged 0.96 ± 0.49, with coarse sand sediment (0.7 ± 0.25 mm) (Table 1).

**Telchac Reef.** The Telchac rocky reef (Fig 4a in S3 Fig) is located 13.14 km from Telchac port, this reef covers a 500 m (E-W) x 300 m (N-S) polygon with 26 E-W transects. It exhibits irregular rocky ridges up to 1.5 m high, with steps running N-S and NW-SE (Fig 4b). Depths range from −12.69 m to −11.08 m, with deeper central areas forming caves (Fig 4c). The R.I. averaged 0.95 ± 0.14, with fine sand sediment (0.132 ± 0.05 mm) (Table 1).

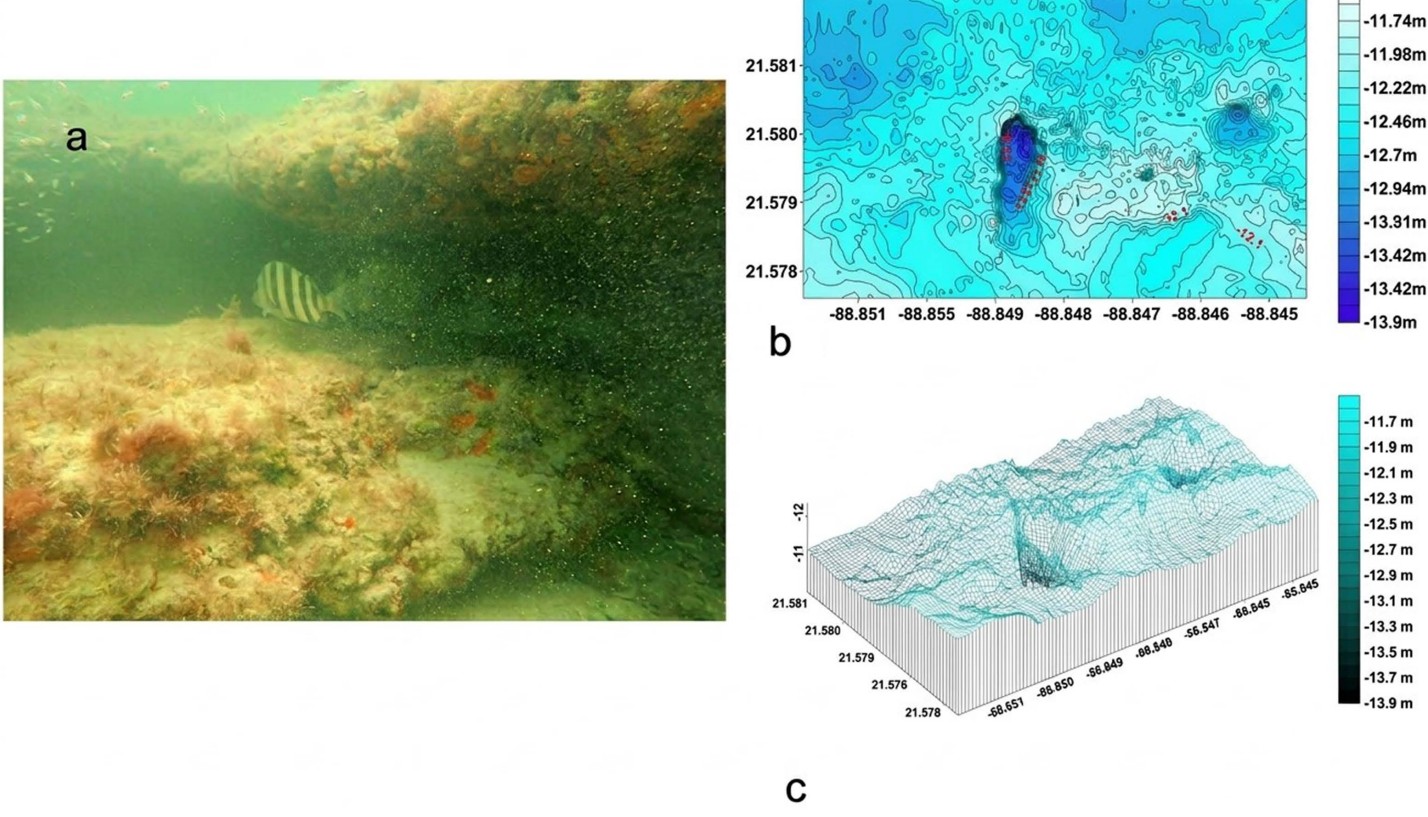

**Fig 2. a) Photograph of the habitat of la Poza rocky reef. Fig 2b): Bathymetric map of La Poza reef, Fig 2c) 3D model of La Poza, rocky structure and cave systems.**

**Table 1. Environmental factors and characteristics of the substrate sediment across sites.**

|  | LSAT Sediment depth (mm) | R.I. | Grain size (mm) | Type of sediment | Depth (m) | Sal PSU | Temp °C | pH |
|---|---|---|---|---|---|---|---|---|
| Telchac | 0.58±0.23 | 0.95±0.14 | 0.13±0.053 | fine sand | 11.88±1.62 | 34 | 26.8 | 8.14 |
| Dzilam La poza | 2.17±0.88 | 0.91±0.35 | 0.72±0.22 | Coarse sand | 12.75±2.3 | 33 | 28.3 | 8.04 |
| Dzilam SMR | 0.90±0.50 | 0.96±0.49 | 0.70±0.25 | Coarse sand | 5.68±1.25 | 33 | 28.9 | 8.04 |
| Progreso | 1.13±0.68 | 0.93±0.11 | 0.33±0.11 | Middle sand | 14.78±1.49 | 32 | 28.9 | 8.05 |
| Chicxulub | 0.74±0.46 | 0.92±0.45 | 0.18±0.09 | fine sand | 11.89±1.61 | 33.7 | 27.3 | 8.05 |

values are presented as Mean±SD. SMR: Small Mountain Range, R.I.: Rugosity index, Sal: Salinity, Temp: Temperature.

**Progreso Reef.** The Progreso rocky reef (Fig 5a in S4 Fig) is situated 17.4 km from Progreso city, this reef spans a 700 m (N-S) x 600 m (E-W) polygon with 37 E-W transects. Ridges and caves cover 80% of the area, with depths from −15.55 m to −14.02 m and elevations up to 1.5 m (Fig 5b). The structure tapers southeastward (Fig 5c). The R.I. averaged 0.931±0.11, with medium coarse sand (0.337±0.11 mm) (Table 1).

**Chicxulub Reef.** The Chicxulub rocky reef (Fig 6a in S5 Fig) is located 15.7 km from Chicxulub port, this reef covers a 200 m (E-W) x 100 m (N-S) polygon with 10 transects. A 150 m-long ridge (NW-SE) rises 1.5 m, with depths from −12.69 m to −11.08 m (Fig 6b). Central deep areas form ridges and caves, with potential southern extension (Fig 6c). The R.I. averaged 0.922±0.45, with fine sand sediment (0.182±0.092 mm) (Table 1).

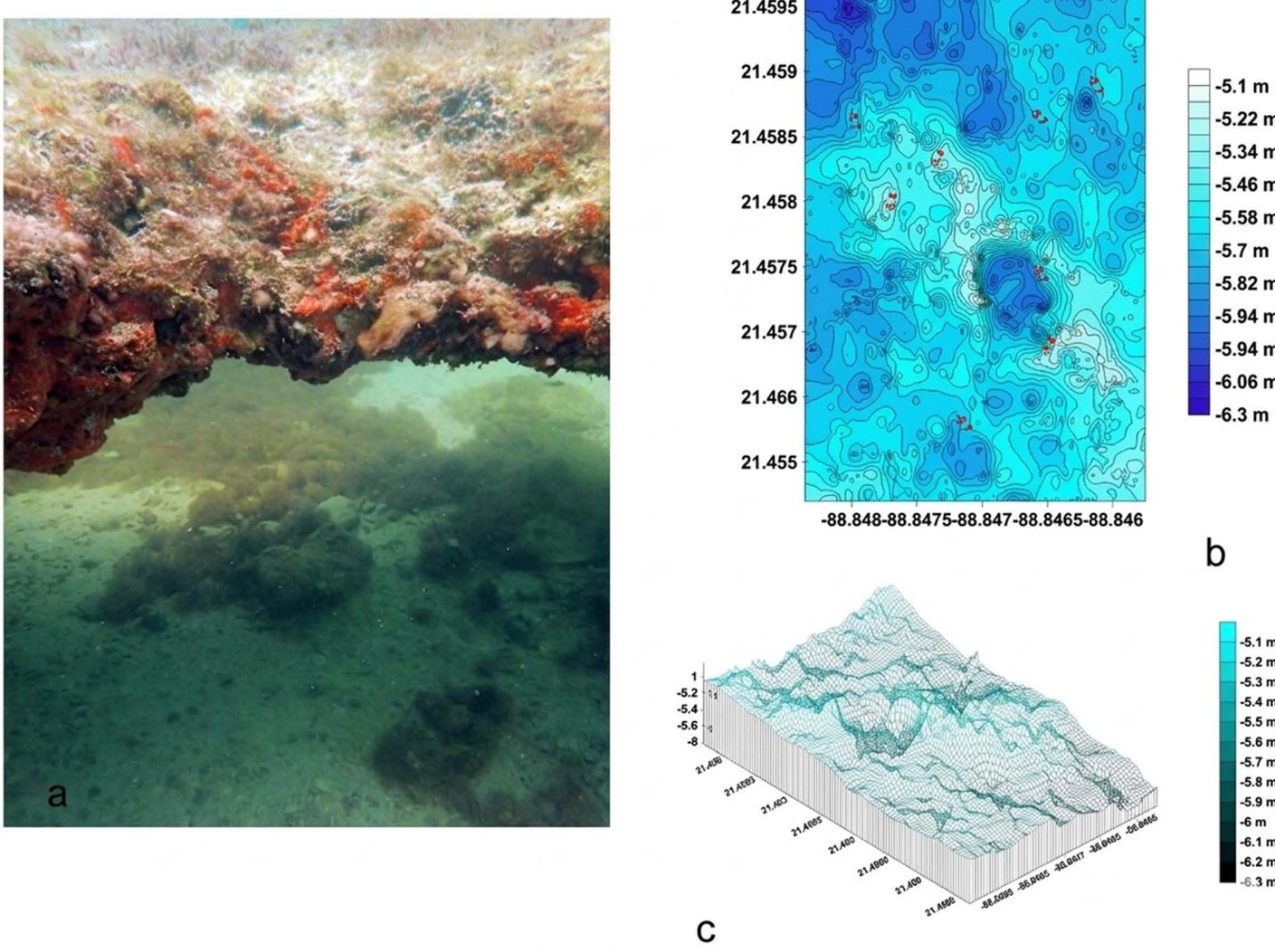

**Fig 3. a) Photograph of the habitat of SMR rocky reef. Fig 3b): Bathymetric map of SMR. Fig 3c) 3D model of SMR.**

## Benthic community composition

**Seasonal and site-specific patterns.** Macroalgae coverage is a dominant component across all locations but varies in proportion. The LSAT proportion also varies considerably, suggesting significant differences in seabed characteristics. The presence of rock/sand is relatively constant, albeit with slight variations. The presence of sessile invertebrates is generally low (Fig 7 in S7 Fig).

The benthic composition percentages showed that Telchac had the highest macroalgal cover (78.6% rainy, 79.3% dry) and lowest LSAT (15.7% rainy, 4.6% dry). Progreso showed high LSAT (51.7% rainy) and reduced macroalgae (31.8% rainy). Chicxulub had high macroalgae (83.6% rainy) and low LSAT (2.1% rainy). Dzilam SMR and La Poza exhibited moderate LSAT and macroalgal cover, varying seasonally. (Fig 7).

Macroalgae dominated benthic cover across all sites (62%), featuring genera such as *Alsidium, Halymenia, Dictyota, Halimeda, and Caulerpa* (Fig 8a,b) LSAT covered 21% (Fig 8c in S8 Fig.), while sessile invertebrates (primarily sponges and small coral) (Fig 8d,e). were less abundant, and the substrate was formed by sand or rock (Fig 8f).

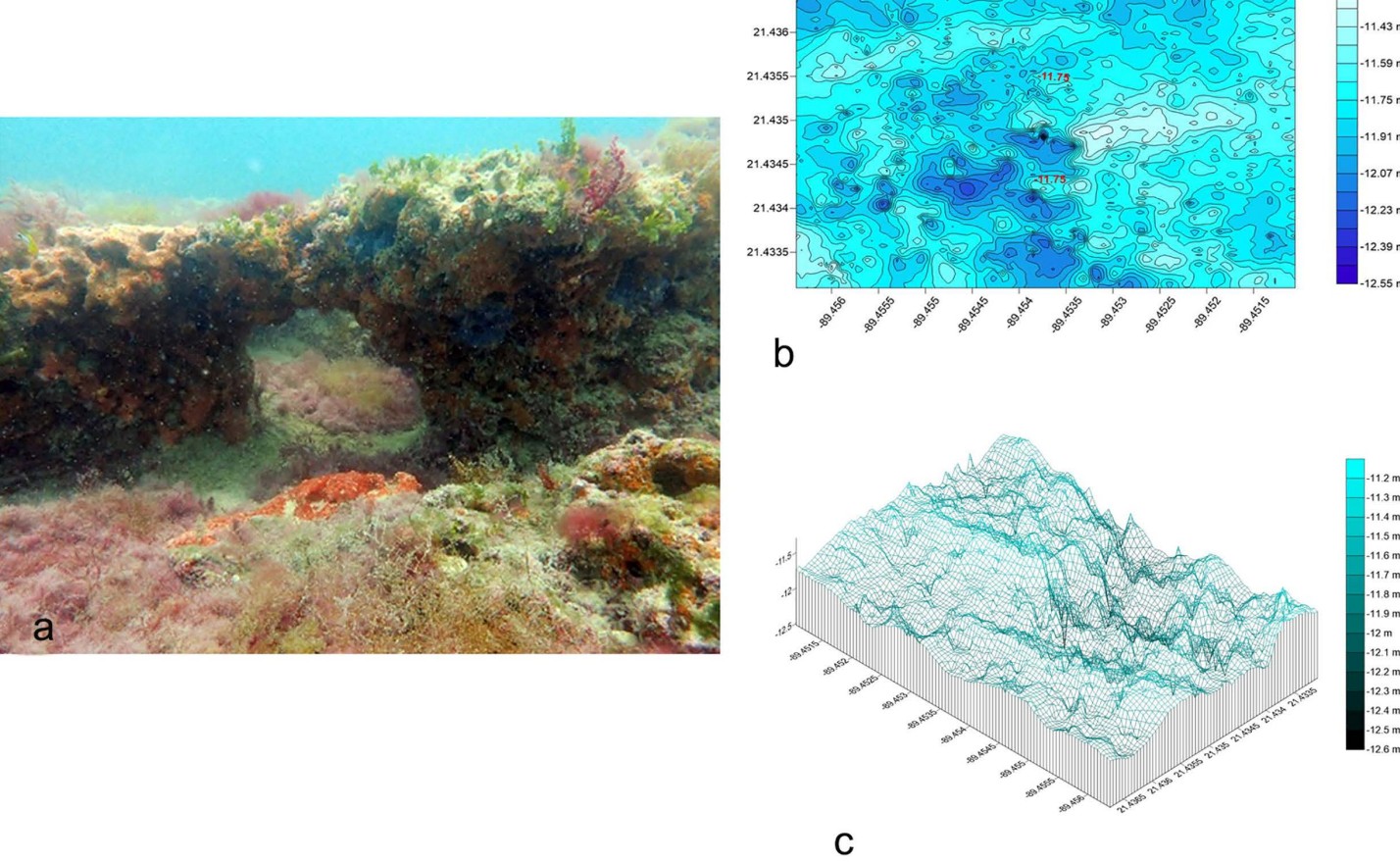

**Fig 4. a) Photograph of the habitat of Telchac rocky reef. Fig 4b) Bathymetric map of Telchac reef. Fig 4c) 3D model of Telchac reef, depicting cave systems and southward extension.**

**Statistical analysis.** Benthic cover, structural complexity, and sediment depth varied markedly among reefs and between seasons, with strong evidence of site-specific seasonal dynamics (Table 2). Macroalgae dominated most reef substrates (mean ± SE: 65.7 ± 7.5%), followed by live coral (LSAT: 22.5 ± 5.6%), sessile invertebrates (4.3 ± 1.2%), rock/sand (7.8 ± 1.6%), and other biota (5.0 ± 2.0%). Rugosity was uniformly high (0.93 ± 0.01), indicating complex reef topography across all sites

Although not statistically significant (α = 0.05), the two-way ANOVA revealed a notable interaction between reef site and season for macroalgal cover ($F_{(3,2)}$ = 4.99, p = 0.08, $\eta^2$ = 0.88; Table 3), suggesting that seasonal trends in macroalgal abundance vary across locations. These site and season-specific differences in benthic community composition are illustrated in Fig 7, which shows mean percentage cover (± SE) for the main benthic categories. Macroalgal dominance was evident across most site season combinations. Telchac exhibited the highest macroalgal abundance during the dry season (94.4%), whereas Progreso showed a peak in the rainy season (31.8% rainy vs. 74.9% dry). Dzilam presented intermediate values, with a slight increase in macroalgal cover during the dry season.

Sessile invertebrate cover did not differ significantly across reef sites or seasons (all p > 0.20), although Progreso during the dry season showed relatively elevated values (12.7%).

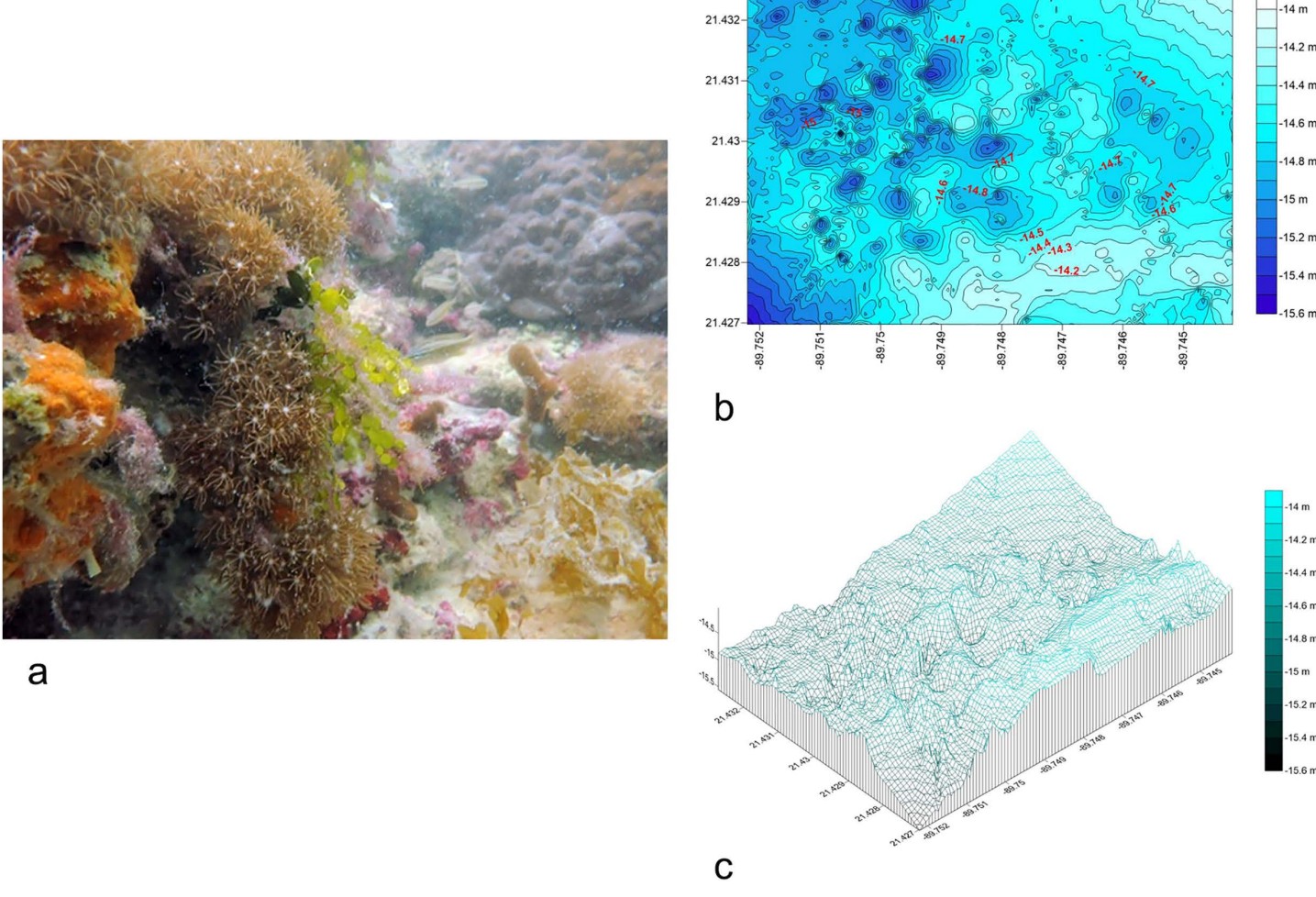

**Fig 5.  a) Photograph of the habitat of Progreso rocky reef. 5b) Bathymetric map of Progreso reef. 5c) 3D model of Progreso reef.**

Sediment depth varied significantly among reef sites ($F_{(3,2)} = 9.87$, $p = 0.03$, $\eta^2 = 0.94$), primarily due to Dzilam exhibiting the deepest sediment layers (mean 1.78 mm), compared to less than 1.4 mm at other sites. No significant differences were detected for rugosity or reef depth ($p > 0.45$).

The data revealed divergent seasonal trends in macroalgal cover across the reef sites. Telchac and Dzilam showed higher macroalgal cover during the dry season (Telchac: +15.7% relative to rainy season; Dzilam: +17.8% relative to rainy season), whereas Progreso exhibited substantially greater macroalgal cover in the rainy season (+43.1% relative to dry season).

The analysis of the linear regression relationship between the percentage of macroalgal abundance and the LSAT across the Yucatan rocky reefs shows a strong negative correlation ($r = -0.754725$). The statistical significance ($P < 0.0001$) underscores a robust inverse relationship (Table 4).

The relation between R.I. and macroalgal abundance shows a moderate positive relationship ($r = 0.254088$); suggesting that macroalgal cover increases with greater topographic complexity and that topographic complexity supports macroalgae. The significant P-value (0.0004) confirms the relationship, (Table 2).

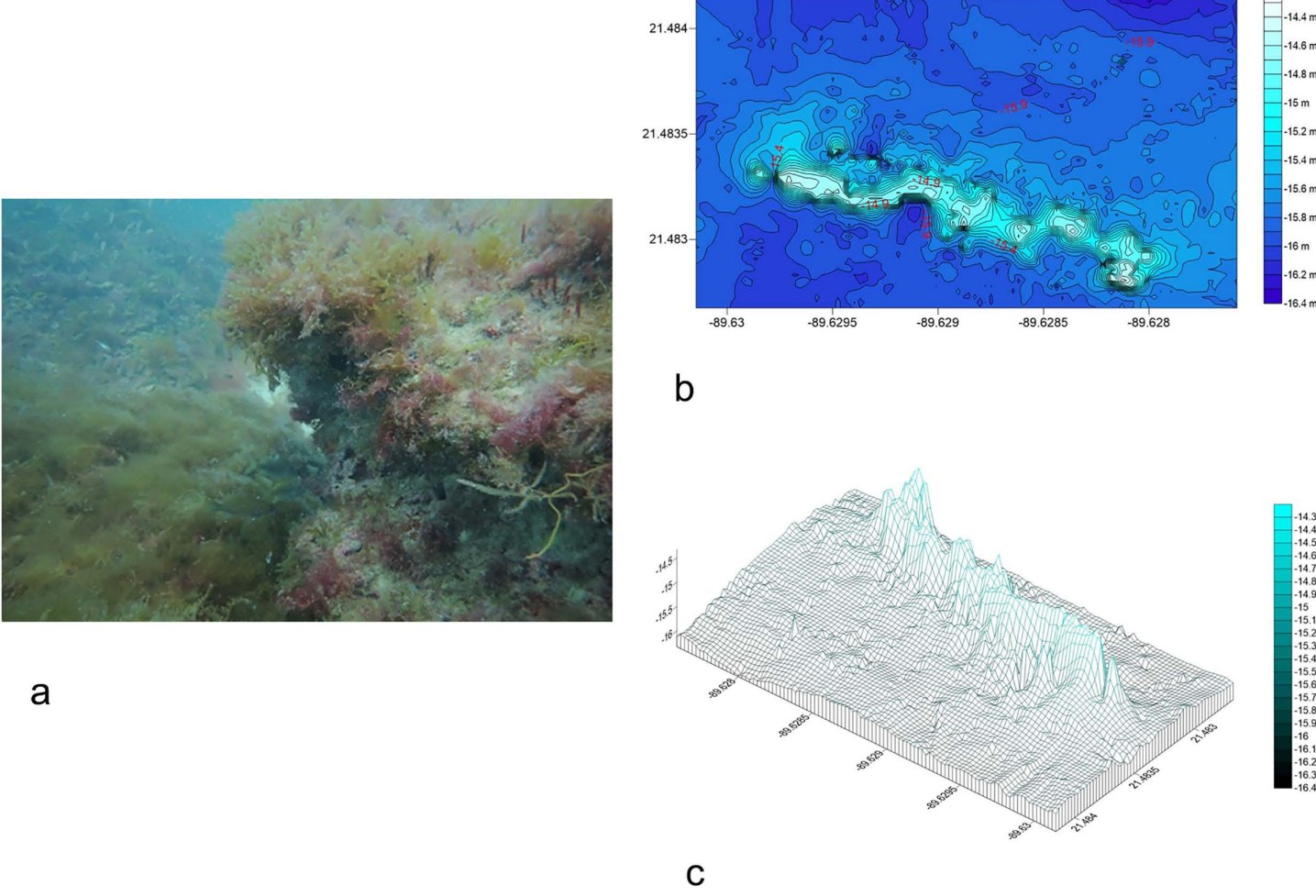

**Fig 6. a) Photograph of the habitat of Chicxulub rocky reef. Fig 6b): Bathymetric map of Chicxulub reef. Fig 6c) 3D model of Chicxulub reef.**

A moderate negative correlation was found between macroalgae and sessile invertebrates ($r = -0.316$, $P < 0.0001$) (Table 4), with ANOVA confirming significance.

To identify the environmental variables that influence the abundance of macroalgae, we fitted a multiple linear regression model using macroalgae cover as the response variable and depth, rugosity index, sediment depth, LSAT cover, season, and reef site as predictors. The model accounted for a substantial proportion of the variance in macroalgal cover (adjusted $R^2 = 0.71$), indicating a strong overall fit. Among the predictors, only LSAT cover emerged as marginally significant ($\beta = -0.61 \pm 0.19$, $t = -3.21$, $p = 0.08$), suggesting that higher LSAT presence is associated with reduced macroalgal cover. All other predictors, including depth and rugosity, were not statistically significant ($p > 0.22$), indicating limited individual explanatory power within the model.

## Discussion

The rocky reefs identified along the northern Yucatan coast challenge prior assumptions of a predominantly sandy seabed [14,15]. Their presence between the 10–15 m isobaths suggests a unique ecological role, potentially as biodiversity reservoirs and connectivity corridors between the Campeche Bank and Caribbean reefs [42,43]. The structural complexity

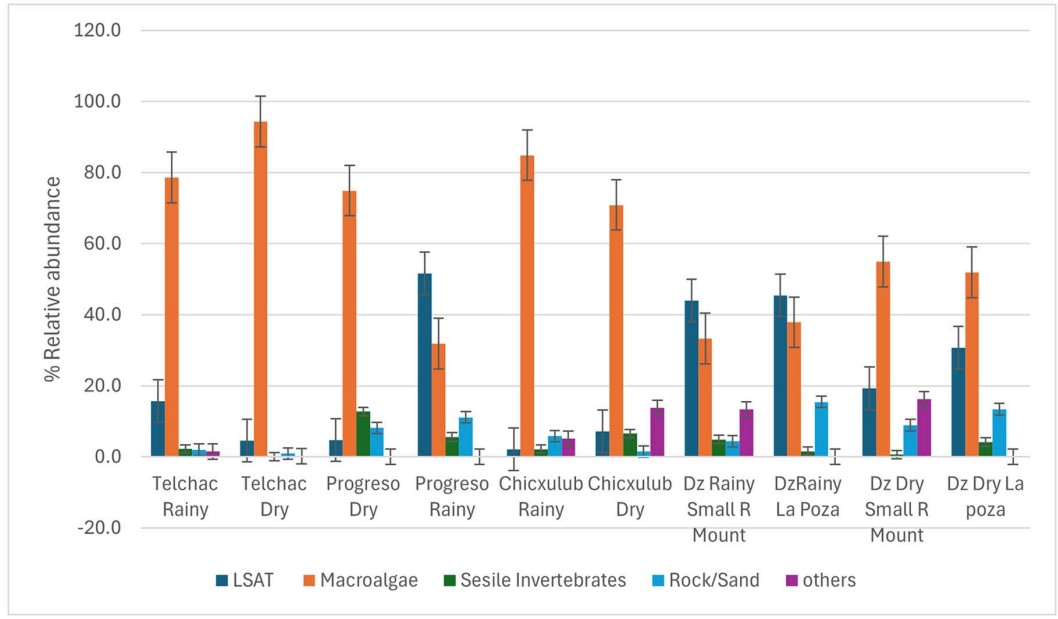

**Fig 7. Mean percentage cover (± standard error) of major benthic categories (macroalgae, LSAT, sessile invertebrates, and bare substrate) at each reef site during the rainy and dry seasons.**

observed as steps, caves, and elevated ridges aligns with patterns known to enhance habitat heterogeneity and species diversity [44].

Ecologically, these reefs likely support diverse communities, as seen in other rocky systems [1,2], enhancing regional resilience [45]. Their study across seasons will clarify temporal dynamics, critical for management [46].

## Topographic characterization of Yucatan rocky reefs

The bathymetric surveys of Yucatan rocky reefs reveal diverse topography influencing ecological dynamics, with sediment grain size varying from coarse sand at Dzilam to fine sand at Telchac and Chicxulub, likely shaping benthic communities [47]. The rugosity also may facilitate macroalgal attachment, giving substrate complexity drives algal community structure [48]. These topographic and sedimentological contrasts suggest that reef morphology and sediment dynamics are critical drivers of ecological patterns, warranting further investigation into their roles in benthic resilience [45].

Dzilam stands out for its elevated sediment depth (mean 1.78 mm; reef effect: $F_{(3,2)} = 9.87$, $p = 0.03$), particularly at the La Poza sub-site (3.0 mm rainy season). Despite shallow depths (5.68–12.75 m), macroalgae cover remained moderate (35.6–53.4%), cave-rich structure enhances habitat diversity [44],., while SMR's shallower slabs may boost light for macroalgae possibly due to light attenuation by suspended sediments or physical disturbance from swell exposure [48].Telchac's ridges and fine sand reduce sedimentation stress, favoring macroalgae [37,49]. Progreso's ridge-cave system with medium coarse sand indicates moderate energy [47], correlating granulometry with benthic assemblages. The Chicxulub's ridge with fine sand and caves reflects moderate complexity. This reef's ecological role may parallel Telchac's, though its smaller size suggests localized significance.

Although 3D bathymetric models were generated for visualization, they lacked the spatial resolution and mesh fidelity required to extract quantitative rugosity metrics. Therefore, we were unable to derive multi-scale rugosity values from these models. We recommend future studies incorporate high-resolution 3D mapping to enhance topographic characterization. While this study focused on the ecological and topographic characterization of rocky reefs, the geological origin of

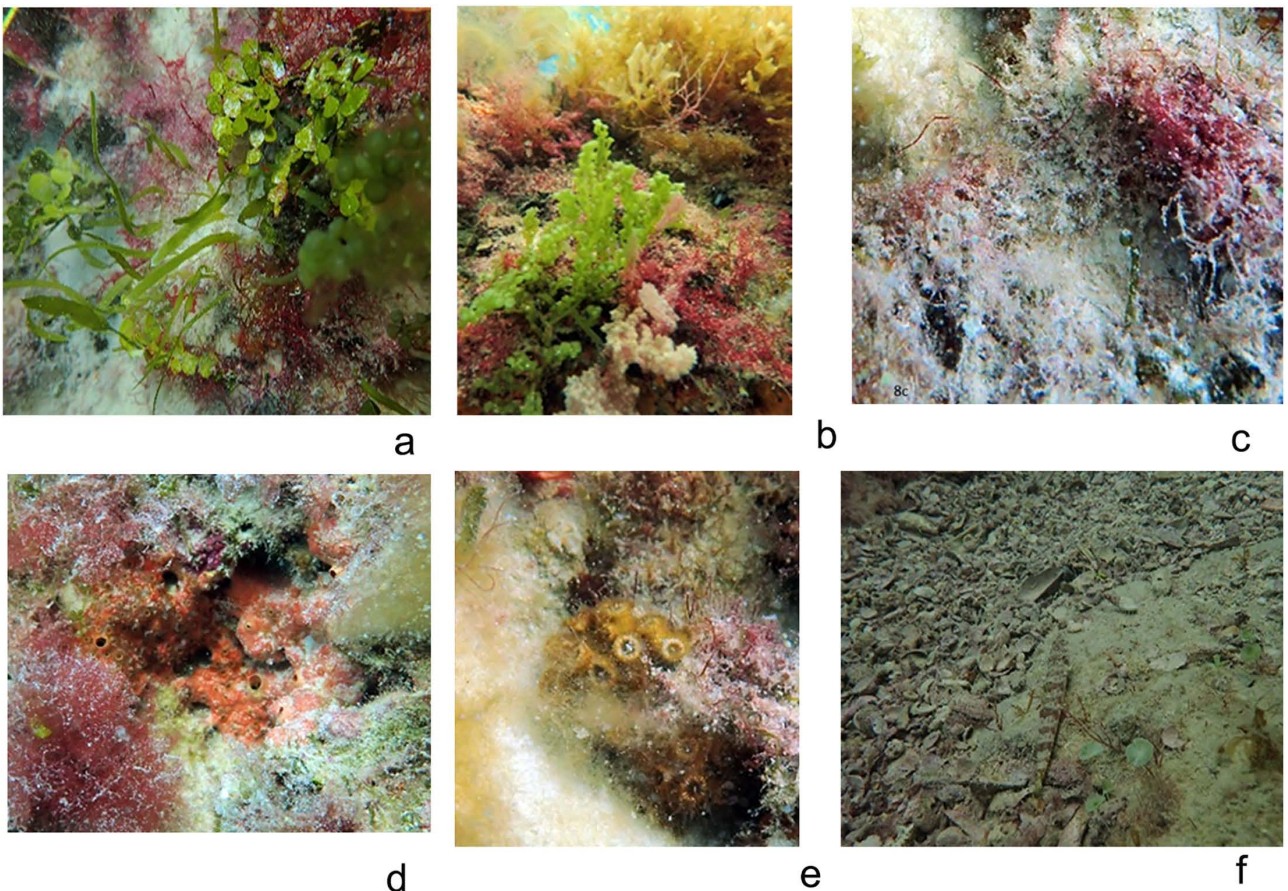

**Fig 8. Benthic composition, a) Macroalgae with *Halimeda* sp, *Caulerpa* sp and Rhodophytes as *Alsidium* sp, b) macroalgae with *Dictyota* sp., *Padina* sp., *Caulerpa* sp. and Rhodophytes. c)LSAT, d) Sponges, e) Corals, f) Rock/sand substrate.**

**Table 2. Mean benthic cover by reef and season.**

| Reef | Season | LSAT (%) | Macroalgae (%) | Sessile Invert. (%) | Rock/Sand (%) | Others (%) |
|---|---|---|---|---|---|---|
| Telchac | Rainy | 15.7 | 78.6 | 2.2 | 2.0 | 1.5 |
|  | Dry | 4.6 | 94.4 | 0.0 | 0.9 | 0.2 |
| Progreso | Dry | 4.7 | 74.9 | 12.7 | 8.1 | 0.0 |
|  | Rainy | 51.7 | 31.8 | 5.6 | 11.1 | 0.0 |
| Chicxulub | Rainy | 2.1 | 84.9 | 2.2 | 5.8 | 5.1 |
|  | Dry | 7.2 | 70.9 | 6.6 | 1.5 | 13.8 |
| Dzilam SRM | Rainy | 45.4 | 37.86 | 1.6 | 15.43 | 0 |
|  | Dry | 30.7 | 51.92 | 4.2 | 13.38 | 0 |
| Dzilam La Poza | Rainy | 19.2 | 54.91 | 0.6 | 8.93 | 16.2 |
|  | Dry | 44.0 | 33.33 | 4.9 | 4.39 | 13.4 |

these formations remains an open question. Given the proximity of the study sites to the Chicxulub impact structure—a 180 km-wide crater formed by a Cretaceous–Paleogene asteroid impact—there is a compelling need for dedicated geological surveys to determine whether these reefs are remnants of impact-related uplift, karstic exposure, or other tectonic

**Table 3. Two-way ANOVA results.**

| Variable | Source | df | F | p |
|---|---|---|---|---|
| Macroalgae | Reef | 3 | 4.98 | 0.08 |
| | Season | 1 | 0.40 | 0.56 |
| | Reef×Season | 3 | 4.99 | 0.08 |
| LSAT | Reef | 3 | 12.41 | 0.025 |
| | Season | 1 | 0.69 | 0.456 |
| | Reef×Season | 3 | 5.26 | 0.046 |
| Sessile Invertebrates | Reef | 3 | 1.85 | 0.29 |
| | Season | 1 | 3.43 | 0.21 |
| | Interaction | 3 | 2.62 | 0.24 |
| Sediment Depth | Reef | 3 | 9.87 | 0.03 |
| | Season | 1 | 0.45 | 0.57 |
| | Interaction | 3 | 1.12 | 0.51 |
| Rugosity | Reef | 3 | 1.23 | 0.47 |
| | Season | 1 | 0.01 | 0.93 |

**Table 4. Correlation matrix of benthic composition and substrate variables R.I.: Rugosity index. \*\*. Correlation is significant at the 0.01 level (2-tailed). \*. Correlation is significant at the 0.05 level (2-tailed).**

| | Macroalgae % | LSAT % | Sessile Invertebrate % | R.I. | Sediment depth mm | Rock/sand % |
|---|---|---|---|---|---|---|
| Macroalgae % | 1 | −0.754** | −0.315* | 0.254* | −0.243* | −0.383** |
| LSAT % | −0.754** | 1 | −0.070 | −0.159* | 0.404** | 0.110 |
| Sessile Invertebrate % | −0.315* | −0.070 | 1 | −0.057 | −0.141 | −0.035 |
| R.I | 0.254* | −0.159* | −0.057 | 1 | −0.066 | 0.077 |
| Sediment depth mm | −0.243* | 0.404** | −0.141 | −0.066 | 1 | 0.086 |
| Rock/sediment % | −0.383** | 0.110 | −0.035 | 0.077 | 0.086 | 1 |

processes. Previous studies have documented the influence of the Chicxulub crater on regional geomorphology, including the formation of cenote rings and subsurface structural features [9–13]. However, the lithological composition, diagenetic history, and potential hydrothermal alterations of these reef substrates remain unexplored. Integrating petrographic, geochemical, and seismic analyses in future research will be essential to elucidate the genesis of these rocky reefs and their relationship to the broader geological evolution of the Yucatan Platform., we assume that this could explain this morphology, contrasting with sandy seabed assumptions [14,15].

## Benthic Community Composition

The benthic communities of Yucatán's rocky reefs exhibit pronounced spatial and seasonal heterogeneity, driven by site-specific responses to seasonal forcing rather than uniform regional patterns, the dominance of macroalgae aligns with regional studies [23,50], documenting macroalgal prevalence on Yucatan reefs. This pattern, consistent across sites, reflects the ecological cornerstone role of macroalgae [51]. Sponges, the primary invertebrate, indicate a secondary but significant benthic component [20,52], while rock/sand fluctuations suggest sediment dynamics as a limiting factor [53].

Our findings align with patterns observed in other rocky reef systems globally, reinforcing the ecological significance of these habitats. For instance, rocky reefs in the Gulf of California support distinct cryptobenthic fish communities compared to adjacent coral habitats, underscoring the need for habitat-specific conservation strategies [54]. Similarly, studies in La Paz Bay demonstrate how anthropogenic pressures such as fishing and coastal development can significantly alter

reef-associated fish assemblages, highlighting the vulnerability of these ecosystems to human impacts [55]. In the Pacific coast of Colombia, rocky reefs have been characterized as biodiversity hotspots, yet remain underrepresented in regional conservation planning, mirroring the research gaps we identify for the Yucatan coast [1].

Benthic composition highlights macroalgal diversity and seasonal dynamics. Telchac's high macroalgal cover and low LSAT suggest minimal competition, consistent with studies attributing macroalgal dominance to spatiotemporal resource availability [56]. Dzilam La Poza's substantial macroalgal presence with moderate LSAT contrasts with Progreso's elevated LSAT and reduced macroalgae, supporting sedimentation's role in altering rocky coast assemblages [57]. Chicxulub's consistent macroalgal dominance and low LSAT suggest stability, while Dzilam's moderate variability aligns with seasonal nutrient shifts [58]. However, site-specific and seasonal variations highlight the influence of environmental drivers such as turbidity, nutrient inputs, and sediment dynamics [41,49,57]. Moreover, research from the East Sea of South Korea shows that benthic community composition is strongly influenced by substrate type and depth gradients [2], a dynamic consistent with our observations of macroalgae-LSAT interactions and the role of sediment grain size and rugosity in shaping benthic structure. These comparisons emphasize the broader relevance of our study and the urgent need to integrate rocky reefs into marine spatial planning and biodiversity assessments.

The observed negative association between macroalgal cover and LSAT suggests competitive exclusion, with macroalgae declining as LSAT increases. This pattern aligns with studies noting algal turf suppression by sessile taxa [59] and linking sediment-laden turfs to reduced settlement success [60]. The model's explanatory power suggests LSAT as a primary limiter of macroalgal expansion, consistent with environmental controls of benthic cover [61].

Even though topographic complexity enhances habitat diversity, supporting macroalgal attachment and benthic richness [37,48], the weak positive relationship with rugosity implies that topographic complexity modestly enhances macroalgal presence, but unmodeled factors, such as herbivory [62] or sediment dynamics [63], may confound this relationship. Sediment grain size variations, from fine sand to coarse sand, modulate these patterns, with finer sediments correlating with higher macroalgal cover and coarser sediments linked to increased LSAT presence [47].

The negative but weak correlation with sessile invertebrates reinforces competition, though its limited explanatory power and serial correlation indicate ecological complexity beyond simple substrate rivalry. Sediments and herbivory are sensitive indicators of reef degradation, suggesting broader biotic interactions at play [64].

Sediment and rugosity characteristics reinforce these findings, with Telchac's fine sand and high R.I. favoring macroalgae, while Dzilam and Progreso's coarser sediments and moderate R.I.s support higher LSAT. Benthic composition data confirms macroalgal resilience across seasons, yet LSAT peaks signal potential vulnerability to phase shifts [45].

The rocky reefs documented in this study are ecologically significant but remain vulnerable to multiple stressors. Coastal infrastructure expansion and sediment runoff threaten reef integrity, while artisanal and industrial fishing exert pressure on associated fauna [65,66]. Climate change compounds these threats by altering thermal regimes and increasing the frequency of extreme weather events, which may shift benthic community dynamics or trigger phase shifts [27,67]. Given these risks, the inclusion of rocky reefs in conservation planning—alongside coral reefs—is imperative. Our findings provide baseline data to support such integration and highlight the urgency of protecting these overlooked ecosystems

## Conclusions

The rocky reefs of the Yucatan Peninsula reveal a dynamic interplay of topography, sedimentology, and biotic interactions, with macroalgae as the prevailing ecological force. Bathymetric surveys show how topographic complexity boosts habitat diversity, giving opportunities for macroalgal attachment and benthic richness. Sediment grain size, ranging from fine to coarse sand, shapes these patterns: finer sediments support greater macroalgal cover, while coarser ones correlate with elevated Long Sediment-Laden Algal Turf (LSAT) presence. Benthic data affirms macroalgae's seasonal resilience, though LSAT spikes suggest susceptibility to ecological phase shifts. Macroalgal dominance persists due to minimal

substrate competition, favorable topography, and adaptability to stable substrates and resource availability, modulated by sediment type and environmental fluctuations. This study is the first approximation to establish a baseline of these rocky reefs and their importance as vital biodiversity hubs, providing critical data and underscoring the need for deeper ecological research into Yucatan's rocky reef systems.

## Supporting information

**S1 Fig. Dzilam La Poza.** 1a) Substrate, LSAT and invertebrates, 1b) Substrate and Corals.
(TIF)

**S2 Fig. Dzilam Small Mountain Range reef.** 2a) Photograph of the topography of SMR. 2b) Substrate LSAT, algae and sponge, 2b).
(TIF)

**S3 Fig. Telchac Reef.** 3a) Photograph of the algae and sponge in Telchac rocky reef. 3b): Substrate, red algae and coral.
(TIF)

**S4 Fig. Progreso reef. 4a) Photograph of substrate, LSAT and corals.** 4b) Red algae and invertebrates of Progreso rocky reef.
(TIF)

**S5 Fig. Chicxulub rocky reef.** 5a) Photograph of Measurements of algae and substrate. 5b) Photograph of the bottom of Chicxulub rocky reef.
(TIF)

## Acknowledgments

This is contribution # 2057from the Institute of Environment at FIU.

## Author contributions

**Conceptualization:** Ileana Ortegón-Aznar, Johnny Omar Valdez-Iuit, Armin N. Tuz-Sulub.

**Data curation:** Ileana Ortegón-Aznar, Alain Duran.

**Formal analysis:** Ileana Ortegón-Aznar.

**Funding acquisition:** Ileana Ortegón-Aznar.

**Investigation:** Ileana Ortegón-Aznar, Johnny Omar Valdez-Iuit, Armin N. Tuz-Sulub.

**Methodology:** Ileana Ortegón-Aznar, Johnny Omar Valdez-Iuit, Armin N. Tuz-Sulub.

**Project administration:** Ileana Ortegón-Aznar.

**Resources:** Ileana Ortegón-Aznar.

**Software:** Johnny Omar Valdez-Iuit.

**Supervision:** Ileana Ortegón-Aznar, Armin N. Tuz-Sulub.

**Validation:** Ileana Ortegón-Aznar.

**Visualization:** Ileana Ortegón-Aznar.

**Writing – original draft:** Ileana Ortegón-Aznar.

**Writing – review & editing:** Johnny Omar Valdez-Iuit, Armin N. Tuz-Sulub.

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
