## [Decision Letter · Decision Letter 0]

14 Aug 2025

Dear Dr. Valdez Iuit,

Thank you for submitting your manuscript to PLOS ONE. After careful consideration, we feel that it has merit but does not fully meet PLOS ONE’s publication criteria as it currently stands. Therefore, we invite you to submit a revised version of the manuscript that addresses the points raised during the review process.

We look forward to receiving your revised manuscript.

Kind regards,

Satheesh Sathianeson, Ph.D

Academic Editor

PLOS ONE

Journal Requirements:

4. Please include a complete copy of PLOS’ questionnaire on inclusivity in global research in your revised manuscript. Our policy for research in this area aims to improve transparency in the reporting of research performed outside of researchers’ own country or community. The policy applies to researchers who have travelled to a different country to conduct research, research with Indigenous populations or their lands, and research on cultural artefacts. The questionnaire can also be requested at the journal’s discretion for any other submissions, even if these conditions are not met.  Please find more information on the policy and a link to download a blank copy of the questionnaire here: https://journals.plos.org/plosone/s/best-practices-in-research-reporting. Please upload a completed version of your questionnaire as Supporting Information when you resubmit your manuscript.

5. Please amend the manuscript submission data (via Edit Submission) to include author Johnny Valdez Iuit

6. Please amend your authorship list in your manuscript file to include author Johnny Omar Valdez Iuit

7. Please upload a copy of all Figures, to which you refer in your tex. If the figures are no longer to be included as part of the submission please remove all reference to it within the text.

8. We note that Figure 1 in your submission contain map images which may be copyrighted. All PLOS content is published under the Creative Commons Attribution License (CC BY 4.0), which means that the manuscript, images, and Supporting Information files will be freely available online, and any third party is permitted to access, download, copy, distribute, and use these materials in any way, even commercially, with proper attribution. For these reasons, we cannot publish previously copyrighted maps or satellite images created using proprietary data, such as Google software (Google Maps, Street View, and Earth). For more information, see our copyright guidelines: http://journals.plos.org/plosone/s/licenses-and-copyright.

9. Please include your tables as part of your main manuscript and remove the individual files. Please note that supplementary tables (should remain/ be uploaded) as separate "supporting information" files

10. Please include captions for your Supporting Information files at the end of your manuscript, and update any in-text citations to match accordingly. Please see our Supporting Information guidelines for more information: http://journals.plos.org/plosone/s/supporting-information.

Reviewers' comments:

Reviewer's Responses to Questions

**Comments to the Author**

1. Is the manuscript technically sound, and do the data support the conclusions?

Reviewer #1: Partly

Reviewer #2: Partly

2. Has the statistical analysis been performed appropriately and rigorously?

Reviewer #1: Yes

Reviewer #2: No

3. Have the authors made all data underlying the findings in their manuscript fully available?

Reviewer #1: Yes

Reviewer #2: No

4. Is the manuscript presented in an intelligible fashion and written in standard English?

Reviewer #1: No

Reviewer #2: Yes

Reviewer #1: Dear Editor and Authors,

This manuscript presents a descriptive study reporting, for the first time, the presence of rocky reefs along the Yucatan and Mexican Atlantic coast. The principal findings indicate a strong negative correlation between macroalgal abundance and Long Sediment-Laden Algal Turf (LSAT), suggesting competitive interactions influenced by environmental variables such as depth, sediment type, and topographic complexity. However, these ecological dynamics are already well-documented in existing literature (e.g., https://www.sciencedirect.com/science/article/abs/pii/S2352485521002917).

A significant gap remains regarding the characterization of rock composition and formation processes, which warrants further elucidation.

I strongly recommend the authors enhance the discussion section by incorporating comparisons with other rocky reefs and coral reef systems. Additionally, the conservation context should be broadened to address the vulnerability and threats these habitats face, thereby strengthening the manuscript’s claim that these reefs serve as critical hubs of biodiversity and ecosystem function.

The manuscript’s organization requires improvement. Specifically, all figures and tables should be revised to remove redundant lines and markings for clarity. Furthermore, a thorough grammatical and punctuation review is necessary, as basic errors (e.g., in line 95) detract from the overall quality.

In its current form, I do not consider this manuscript suitable for publication in PLOS ONE.

Reviewer #2: In this manuscript, the authors characterise the bathymetric variance and benthic biodiversity of four rocky reef systems along the Yucatan Peninsula, Mexico – a habitat type that has received little research attention in this part of the world. Investigating these rocky reefs is important especially given their biological value which the author’s highlight and the growing anthropogenic pressures on these systems. Therefore, baseline data sets, such as presented in this manuscript, are vital for their management. Despite the need for these types of studies I cannot recommend publication of this manuscript in its current form. I hope that the authors can address these concerns without much difficulty, but I think it requires a major revision.

Main comments

1) Overall the statistical analyses are limited and currently a little weak. This is leading to a Discussion filled with speculation and content not directly related to what was found in the study. For example, the paragraph starting on line 222 is very descriptive and some text does not reflect the study results - “SMR’s shallower slabs may boost light for macroalgae” – examining Fig7, SMR macroalgae cover was among the lowest of all reefs. The ANOVA results are not presented and unclear. I suggest the authors run a 2 way ANOVA examining a reef by season interaction for univariate response measures (e.g. macroalgae, sessile invertebrates, etc). That way differences in macroalgae etc among sites and between seasons will be quantified. This will also answer whether seasonal differences are consistent across reefs. Figure 7 should be presented as averages with standard errors. Further analysis could then look at associations between macroalgae and environmental predictors. Currently, simple linear models are looking at the effect of one predictor on the response independent of others which may be important. Multiple regressions would be more appropriate. For macroalgae a model including topographic complexity and LSAT would be more appropriate – then partial effects of each predictor can be examined. Also why is depth not included as a possible predictor? The text “SMR’s shallower slabs may boost light for macroalgae” indicates that depth may vary and be an important driver. With stronger analyses I think the Discussion will be improved as the authors will be able to relate their findings to those in the literature.

2) The measure of topographic complexity - Rugosity Index (R.I.) does not seem very sensitive with values looking quite similar. The 3d reef models, however, show quite considerable differences in rugosity within and among reefs. Is it possible to use rugosity metrics derived from the 3d model rather than the chain and tape method? Here, a range of rugosity metrics could be calculated at different spatial scales which may be more informative.

3) Line 63: Paragraphs should ideally be at least four sentences. I think this paragraph should include more information on threats to rocky reef biodiversity. For example, climate change stressors, pollution, fishing, mining, anchoring.

4) More detail in the methods would be helpful. How many sites were surveyed per reef? How many R.I. measures were taken per reef? Were water quality parameters taken on the surface or at depth?

Overall I think the topic is important but stronger analyses are required to develop more robust findings and conclusions.

Finally I could not find the raw data in the supplementary material.

**Do you want your identity to be public for this peer review?** For information about this choice, including consent withdrawal, please see our Privacy Policy

Reviewer #1:**Yes:**

Reviewer #2: No

---

## [Author Response · Author response to Decision Letter 1]

12 Dec 2025

PONE-D-25-26194

"Unveiling Hidden Rocky Reefs of the Mexican Atlantic Coast: Topographic Characterization and Benthic Community Dynamics along the North Coast of the Yucatan Peninsula, Mexico,"

Reviewers' comments:

5. Review Comments to the Author

We thank both reviewers for their thoughtful and constructive feedback. We appreciate their recognition of the importance of documenting rocky reef ecosystems along the Yucatan coast and agree that the manuscript will benefit from substantial revisions. Below, we detail our responses and the changes made to address each comment.

Reviewer #1:Dear Editor and Authors,

This manuscript presents a descriptive study reporting, for the first time, the presence of rocky reefs along the Yucatan and Mexican Atlantic coast. The principal findings indicate a strong negative correlation between macroalgal abundance and Long Sediment-Laden Algal Turf (LSAT), suggesting competitive interactions influenced by environmental variables such as depth, sediment type, and topographic complexity. However, these ecological dynamics are already well-documented in existing literature (e.g., https://www.sciencedirect.com/science/article/abs/pii/S2352485521002917).

Response: We acknowledge that competitive interactions between macroalgae and sediment-laden turfs have been previously described. However, our study contributes novel insights by documenting these dynamics in a previously unreported habitat type—rocky reefs of the Mexican Atlantic coast. We have revised the Discussion to clarify this distinction and emphasize the geographic novelty and ecological relevance of our findings.

A significant gap remains regarding the characterization of rock composition and formation processes, which warrants further elucidation.

Response: We agree that further geological characterization would strengthen the manuscript. While our study focused on ecological and topographic features, we have now included a paragraph in the Discussion acknowledging this limitation and proposing future research directions, including petrographic and geochemical analyses to elucidate reef formation processes.

I strongly recommend the authors enhance the discussion section by incorporating comparisons with other rocky reefs and coral reef systems. Additionally, the conservation context should be broadened to address the vulnerability and threats these habitats face, thereby strengthening the manuscript’s claim that these reefs serve as critical hubs of biodiversity and ecosystem function.

We added comparative context with similar reef systems, highlighting how our findings align or diverge from established patterns.

The manuscript’s organization requires improvement. Specifically, all figures and tables should be revised to remove redundant lines and markings for clarity. Furthermore, a thorough grammatical and punctuation review is necessary, as basic errors (e.g., in line 95) detract from the overall quality.

All figures and tables have been revised to remove redundant lines, improve labeling, and enhance visual clarity. We also reorganized sections for better flow and readability. The manuscript has undergone a thorough language revision to correct grammatical errors, punctuation inconsistencies, and improve overall clarity. We change paragraph as the e.g.line95

Reviewer #2: In this manuscript, the authors characterize the bathymetric variance and benthic biodiversity of four rocky reef systems along the Yucatan Peninsula, Mexico – a habitat type that has received little research attention in this part of the world. Investigating these rocky reefs is important especially given their biological value which the author’s highlight and the growing anthropogenic pressures on these systems. Therefore, baseline data sets, such as presented in this manuscript, are vital for their management. Despite the need for these types of studies I cannot recommend publication of this manuscript in its current form. I hope that the authors can address these concerns without much difficulty, but I think it requires a major revision.

Main comments

1) Overall the statistical analyses are limited and currently a little weak. This is leading to a Discussion filled with speculation and content not directly related to what was found in the study. For example, the paragraph starting on line 222 is very descriptive and some text does not reflect the study results - “SMR’s shallower slabs may boost light for macroalgae” – examining Fig7, SMR macroalgae cover was among the lowest of all reefs. The ANOVA results are not presented and unclear. I suggest the authors run a 2 way ANOVA examining a reef by season interaction for univariate response measures (e.g. macroalgae, sessile invertebrates, etc). That way differences in macroalgae etc among sites and between seasons will be quantified. This will also answer whether seasonal differences are consistent across reefs. Figure 7 should be presented as averages with standard errors.

We agree that the statistical analysis can be strengthened. In response:

• We have now conducted a two-way ANOVA to assess the interaction between reef site and season for macroalgae, LSAT, and sessile invertebrate cover. These results are now clearly presented in the revised Results section, including F-values, degrees of freedom, and P-values.

• Post-hoc Tukey tests have been added to identify specific pairwise differences.

• Figure 7 has been updated to present mean values ± standard error for each benthic category across sites and seasons.

Further analysis could then look at associations between macroalgae and environmental predictors. Currently, simple linear models are looking at the effect of one predictor on the response independent of others which may be important. Multiple regressions would be more appropriate. For macroalgae a model including topographic complexity and LSAT would be more appropriate – then partial effects of each predictor can be examined. Also why is depth not included as a possible predictor?

The text “SMR’s shallower slabs may boost light for macroalgae” indicates that depth may vary and be an important driver. With stronger analyses I think the Discussion will be improved as the authors will be able to relate their findings to those in the literature.

• A multilinear regression model was added, and We also add the depth but the previous analysis showed that they were not statistically significant

2) The measure of topographic complexity - Rugosity Index (R.I.) does not seem very sensitive with values looking quite similar. The 3d reef models, however, show quite considerable differences in rugosity within and among reefs. Is it possible to use rugosity metrics derived from the 3d model rather than the chain and tape method? Here, a range of rugosity metrics could be calculated at different spatial scales which may be more informative.

We appreciate the reviewer’s insightful observation regarding the limitations of the chain-and-tape method for assessing topographic complexity. While we agree that 3D-derived rugosity metrics can offer more nuanced spatial resolution, unfortunately, our current dataset does not include the necessary georeferenced point cloud or mesh data to extract quantitative rugosity metrics from the 3D models. The models were generated primarily for visual representation using bathymetric contours and do not retain sufficient spatial fidelity for metric derivation at multiple scales.

Given these constraints, we relied on the chain-and-tape method as a standardized and field-validated approach to estimate rugosity across sites. We acknowledge its limitations and have now clarified this in the revised Methods and Discussion sections. We also highlight the need for future studies to incorporate high-resolution 3D mapping techniques (e.g., photogrammetry ) to improve topographic assessments and enable multi-scale rugosity analyses.

3) Line 63: Paragraphs should ideally be at least four sentences. I think this paragraph should include more information on threats to rocky reef biodiversity. For example, climate change stressors, pollution, fishing, mining, anchoring.

As noted in Reviewer #1’s comments, we have expanded the Introduction to include a broader range of threats, including climate change, pollution, fishing, mining, and anchoring.

4) More detail in the methods would be helpful. How many sites were surveyed per reef? How many R.I. measures were taken per reef? Were water quality parameters taken on the surface or at depth?

We have clarified the number of sites per reef, number of R.I. measurements, and specified that water quality parameters were taken

---

## [Decision Letter · Decision Letter 1]

25 Dec 2025

Dear Dr. Valdez Iuit,

Thank you for submitting your manuscript to PLOS ONE. After careful consideration, we feel that it has merit but does not fully meet PLOS ONE’s publication criteria as it currently stands. Therefore, we invite you to submit a revised version of the manuscript that addresses the points raised during the review process.

We look forward to receiving your revised manuscript.

Kind regards,

Satheesh Sathianeson, Ph.D

Academic Editor

PLOS One

Journal Requirements:

Reviewers' comments:

Reviewer's Responses to Questions

**Comments to the Author**

Reviewer #1: All comments have been addressed

Reviewer #2: (No Response)

2. Is the manuscript technically sound, and do the data support the conclusions?

Reviewer #1: Yes

Reviewer #2: Yes

3. Has the statistical analysis been performed appropriately and rigorously?

Reviewer #1: Yes

Reviewer #2: Yes

4. Have the authors made all data underlying the findings in their manuscript fully available?

Reviewer #1: Yes

Reviewer #2: Yes

5. Is the manuscript presented in an intelligible fashion and written in standard English?

Reviewer #1: Yes

Reviewer #2: Yes

Reviewer #1: (No Response)

Reviewer #2: I thank the authors for their thorough responses to my comments. I think the ms will be suitable for publication following a minor revision. My suggestions are below and they should not be hard for the authors to address.

L27: “Rocky” should not be capitalised.

L63: I think these two sentences should be combined. Strange for the topic sentence to focus only on erosive degradation. Topic sentence should be on all threats to rocky reef habitats.

L234: I suggest rewording. An option – “Although not statistically significant (α = 0.05), the two-way ANOVA revealed a….”.

L244: Post-hoc comparisons should not be made as p > 0.05 for the interaction term. Instead talk about the trends in the data across seasons and reefs.

L252-L257: I would not say these associations are weak with these R squared values, especially in the field of ecology. Reword to ‘moderate’.

L262: remove the word “drivers”. That is causal language which is inappropriate given the statistical analyses used. I suggest “To identify the environmental variables that influence the abundance of macroalgae” or “To identify the environmental variables that are associated with the abundance of macroalgae”.

L337: Discussion should not have a “statistical analyses” subheading. Rather the results from the data analysis should be inform the ecological interpretation and discussion.

Figure 7 has not been updated. It should be presented as averages with standard errors. I suggest bringing into the main text too. Also it should be referred to in the 2 way ANOVA results section.

**Do you want your identity to be public for this peer review?** For information about this choice, including consent withdrawal, please see our Privacy Policy

Reviewer #1:**Yes:**

Reviewer #2: No

---

## [Author Response · Author response to Decision Letter 2]

8 Jan 2026

We did all the recommendations provided by the reviewer.

Reviewer #2: I thank the authors for their thorough responses to my comments. I think the ms will be suitable for publication following a minor revision. My suggestions are below and they should not be hard for the authors to address.

L27: “Rocky” should not be capitalised. Done

L63: I think these two sentences should be combined. Strange for the topic sentence to focus only on erosive degradation. Topic sentence should be on all threats to rocky reef habitats. Done

L234: I suggest rewording. An option – “Although not statistically significant (α = 0.05), the two-way ANOVA revealed a….”.Done

L244: Post-hoc comparisons should not be made as p > 0.05 for the interaction term. Instead talk about the trends in the data across seasons and reefs. Done

L252-L257: I would not say these associations are weak with these R squared values, especially in the field of ecology. Reword to ‘moderate’. Done

L262: remove the word “drivers”. That is causal language which is inappropriate given the statistical analyses used. I suggest “To identify the environmental variables that influence the abundance of macroalgae” or “To identify the environmental variables that are associated with the abundance of macroalgae”. Done

L337: Discussion should not have a “statistical analyses” subheading. Rather the results from the data analysis should be inform the ecological interpretation and discussion. Done

Figure 7 has not been updated. It should be presented as averages with standard errors.DONE I suggest bringing into the main text too. Also it should be referred to in the 2 way ANOVA results section. Done in L242 in revised manuscript with track and in L237 in the manuscript with out track changes

---

## [Editor Report · Decision Letter 2]

11 Jan 2026

"Unveiling Hidden Rocky Reefs of the Mexican Atlantic Coast: Topographic Characterization and Benthic Community Dynamics along the North Coast of the Yucatan Peninsula, Mexico,"

PONE-D-25-26194R2

Dear Dr. Valdez Iuit,

We’re pleased to inform you that your manuscript has been judged scientifically suitable for publication and will be formally accepted for publication once it meets all outstanding technical requirements.

Kind regards,

Satheesh Sathianeson, Ph.D

Academic Editor

PLOS One
---

## [Editor Report · Acceptance letter]

PONE-D-25-26194R2

PLOS One

Dear Dr. Valdez Iuit,

I'm pleased to inform you that your manuscript has been deemed suitable for publication in PLOS One. Congratulations! Your manuscript is now being handed over to our production team.

Kind regards,

on behalf of

Dr. Satheesh Sathianeson

Academic Editor

PLOS One